# THE GUIDE AND THE EXPLORER: SMART AGENTS FOR RESOURCE-LIMITED ITERATED BATCH REINFORCEMENT LEARNING

## ABSTRACT

Iterated (a.k.a growing) batch reinforcement learning (RL) is a growing subfield fueled by the demand from systems engineers for intelligent control solutions that they can apply within their technical and organizational constraints. Model-based RL (MBRL) suits this scenario well for its sample efficiency and modularity. Recent MBRL techniques combine efficient neural system models with classical planning (like model predictive control; MPC). In this paper we add two components to this classical setup. The first is a Dyna-style policy learned on the system model using model-free techniques. We call it the guide since it guides the planner. The second component is the explorer, a strategy to expand the limited knowledge of the guide during planning. Through a rigorous ablation study we show that combination of these two ingredients is crucial for optimal performance and better data efficiency. We apply this approach with an off-policy guide and a heating explorer to improve the state of the art of benchmark systems addressing both discrete and continuous action spaces.

## 1 INTRODUCTION

John is a telecommunication engineer. His day job is to operate a mobile antenna. He has about forty knobs to turn, in principle every five minutes, based on about a hundred external and internal system observables. His goal is to keep some performance indicators within operational limits while optimizing some others. In the evenings John dreams about using reinforcement learning (RL) to help him with his job. He knows that he cannot put an untrusted model-free agent on the antenna control (failures are very costly), but he manages to convince his boss to run live tests a couple of days every month.

John's case is arguably on the R&D table of a lot of engineering companies today. AI adoption is slow, partly because these companies have little experience with AI, but partly also because the algorithms we develop fail to address the constraints and operational requirements of these systems. What are the common attributes of these systems?

- They are physical, not getting faster with time, producing tiny data compared to what model-free RL (MFRL) algorithms require for training.
- System access is limited to a small number of relatively short live tests, each producing logs that can be used to evaluate the current policy and can be fed into the training of the next.
- They are relatively small-dimensional, and system observables were designed to support human control decisions, so there is no need to filter them or to learn representations (with the exception when the engineer uses complex images, e.g., a driver).
- Rewards are non-sparse, performance indicators come continually. Delays are possible but usually not long.

The RL setup that fits this scenario is neither pure offline (batch RL; Levine et al. (2020)), since interacting with the system is possible during multiple live tests, nor pure online, since the policy can only be deployed a limited number of times on the system (Fig 1). After each deployment on the real system, the policy is updated offline with access to all the data collected during the

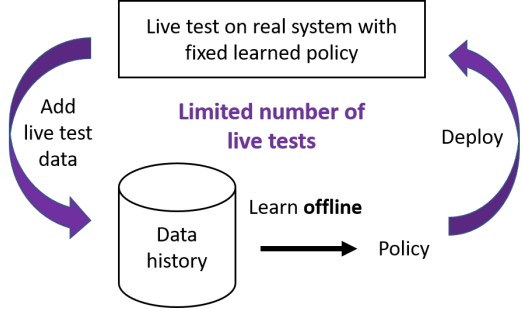

Figure 1: Iterated (a.k.a growing) batch RL. The policy is updated offline between scheduled live tests where the latest learned policy can be deployed on the real system to collect data and further improve itself at the next offline update.

previous deployments, each update benefiting from a larger and more diverse data set. This setup also assumes that the policy cannot be updated online while it is being deployed on the system. We refer to it as *iterated batch RL* (also called *growing batch* (Lange et al., 2012) or *semi-batch* (Singh et al., 1995; Matsushima et al., 2021) in the literature). Furthermore, we are interested in model-based RL (Deisenroth & Rasmussen, 2011; Chua et al., 2018; Moerland et al., 2021). With limited access to the real system, a model of the system transitions can be used to simulate trajectories, either at decision-time to search for the best action (*decision-time planning*, e.g., model predictive control (MPC)) or when learning the policy (*background planning*, e.g., Dyna-style algorithms that learn model-free agents with the model), which makes model-based RL sample efficient (Chua et al., 2018; Wang et al., 2019). Additionally, model-based RL works well on small-dimensional systems with dense rewards, and the system model (data-driven simulator, digital twin) itself is an object of interest because it can ease the adoption of data-driven algorithms by systems engineers.

Given a robust system model, simple model predictive control (MPC) agents using random shooting (RS; Richards (2005); Rao (2010)) or the cross entropy method (CEM; de Boer et al. (2004)) have been shown to perform remarkably well on many benchmark systems (Nagabandi et al., 2018; Chua et al., 2018; Wang et al., 2019; Hafner et al., 2019; Kégl et al., 2021) and real-life domains such as robotics (Yang et al., 2020). However these methods can be time consuming to run at decision-time on the real system, especially when using large models and with the search budgets required for complex action spaces. On the other hand, implementing successfully the seemingly elegant Dyna-style approach (Sutton, 1991; Kurutach et al., 2018; Clavera et al., 2018; Luo et al., 2019), when we learn fast reactive model-free agents on the system model and apply them on the real system, remains challenging especially on systems that require planning with long horizon. Our main findings are that i) the Dyna-style approach can still be an excellent choice when combined with decision-time planning or, looking at it from the opposite direction, ii) the required decision-time planner can be made resource efficient by guiding it with the Dyna-style policy and optionally bootstrapping with its associated value function: this allows an efficient exploration of the action search space (given the limited resource budget) where fewer and shorted rollouts are needed to find the best action to play. We also innovate on the experimental framework (metrics, statistically rigorous measurements), so we can profit from the modularity of the Dyna-style approach by tuning ingredients (model, the MFRL *guide* policy, exploration, planning, bootstrapping; explained in Section 3.2) independently. This modular approach makes engineering easier (as opposed to monolithic approaches like AlphaZero (Silver et al., 2017)), which is an important aspect if we want to give the methodology to non-expert systems engineers.

## 1.1 SUMMARY OF CONTRIBUTIONS

- A conceptual framework with interchangeable algorithmic bricks for iterative batch reinforcement learning, suitable to bring intelligent control into slow, physical, low-dimensional engineering systems and the organizational constraints surrounding them.

- A case study that indicates that the combination of a Dyna-style approach and resource-limited planning can mutually improve each other.

- An ablation study that helped us find the combination of a neural model, a bootstrapping off-policy algorithm guide, and a heating explorer, which brings significant improvement over vanilla agents (MPC, pure Dyna without planning) on both discrete and continuous actions systems.

## 2 RELATED WORK

The MBRL subfield has seen a proliferation of powerful methods, but most of them miss the specific requirements (solving problems irrelevant in this scenario like representation learning or sparse rewards) and missing others (limited and costly system access; time constraints for action search; data taking and experimentation through campaigns, live tests; safety) (Hamrick, 2019).

The Dyna framework developed by Sutton (1991) consists in training an agent from both real experience and from simulations from a system model learned from the real data. Its efficient use of system access makes it a natural candidate for iterated batch RL. The well-known limitation of this approach is the agent overfitting the imperfect system model (Grill et al., 2020). A first solution is to use short rollouts on the model to reduce error accumulation as done in Model-Based Policy Optimization (MBPO; Janner et al. (2019)). Another solution is to rely on ensembling techniques for the model as done in ME-TRPO (Kurutach et al., 2018) and MP-MPO (Clavera et al., 2018). Instead of learning the model and then the policy from the model, Stochastic Lower Bound Optimization (SLBO; Luo et al. (2019)) alternates between model and policy updates. In our ITERATEDMBRL schema (Fig 2), this strategy would couple the LEARN and ACTOR steps, which we do not study in this paper. Finally, Yu et al. (2020) and Kidambi et al. (2020) use a Dyna-style approach in the context of pure batch RL where no further data collection and therefore no further model updates are assumed.

The idea of using a guide and a value function when planning is not novel (Silver et al., 2017; Schrittwieser et al., 2020; Wang & Ba, 2020; Argenson & Dulac-Arnold, 2021; Sikchi et al., 2021). We were greatly inspired by these elements in our objective of building smarter agents as they can make the search more efficient and lead to a better performance. POPLIN-A (Wang & Ba, 2020) relies on behavior cloning (using only real experience, unlike our Dyna-style approach that mainly uses the model), but their decision-time planner is similar to our approach. During the planning, they add a Gaussian noise to the actions recommended by a deterministic policy network and update the noise distribution using a CEM strategy. In a similar way our GUIDE&EXPLORE strategy also adds a carefully controlled amount of noise to the recommended actions. Our results highlight the importance of a well-calibrated exploration. Additionally, our planner does not require to specify the amount of noise beforehand. Argenson & Dulac-Arnold (2021); Lowrey et al. (2019); Sikchi et al. (2021) found that bootstrapping with a value estimate improves the performance of simple guided MPC strategies (see also Bhardwaj et al. (2021)). The popular AlphaZero (Silver et al., 2017) and MuZero (Schrittwieser et al., 2020) algorithms also rely on a guide and a value function for their Monte Carlo Tree Search (MCTS). The principal issue of MuZero (Schrittwieser et al., 2020) in our micro-data iterated batch RL context is that it does not control the number of system access steps: it needs to simulate a lot from the *real* environment to establish the targets for the value function. In these two algorithms the guide is updated from the results obtained during the search that it guided, a procedure known as dual policy Iteration (Anthony et al., 2017; Sun et al., 2018). Furthermore, most of the computation to grow the search tree is done sequentially which results in a slower planner compared to the natural parallelized implementation of our agent. We prefer experimenting with Dyna-style approaches first to leverage popular MFRL algorithms and defer the study of dual policy iteration to future work.

Our results show that decision-time planning is an important ingredient, a claim already made by Hamrick et al. (2021) and Springenberg et al. (2020) among others. Hamrick et al. (2021) use MuZero to run their ablation study while we prefer using an explicit model for practical reasons explained in the introduction. Springenberg et al. (2020) study where additional computation is best spent between policy update and decision-time planning. In our case we are however less concerned by computational resources required to update the policy, as it is done offline, but rather by the time spent at decision-time while interacting with the real system.

## 3 THE FRAMEWORK FOR RESOURCE-LIMITED ITERATIVE BATCH RL

### 3.1 THE FORMAL SETUP

Let $\mathcal{T}_T = \big((\boldsymbol{s}_1, \boldsymbol{a}_1), \ldots, (\boldsymbol{s}_T, \boldsymbol{a}_T)\big)$ be a system trace consisting of $T$ steps of observable-action pairs $(\boldsymbol{s}_t, \boldsymbol{a}_t)$: given an observable $\boldsymbol{s}_t$ of the system state at time $t$, an action $\boldsymbol{a}_t$ was taken, leading to a new system state observed as $\boldsymbol{s}_{t+1}$. The observable vector $\boldsymbol{s}_t = (s_t^1, \ldots, s_t^{d_s})$ contains $d_s$ numerical or categorical variables, measured on the system at time $t$. The action vector $\boldsymbol{a}_t$ contains

$d_a$ numerical or categorical action variables, typically set by a control function $\boldsymbol{a}_t = \pi(\boldsymbol{s}_t)$ of the current observable $\boldsymbol{s}_t$ (or by a stochastic policy $\boldsymbol{a}_t \sim \pi(\boldsymbol{s}_t)$; we will also use the notation $\pi : \boldsymbol{s}_t \rightsquigarrow \boldsymbol{a}_t$). The performance of the policy is measured by the reward $r_t$ which is a function of $\boldsymbol{s}_t$ and $\boldsymbol{a}_t$. Given a trace $\mathcal{T}_T$ and a reward $r_t$ obtained at each step $t$, we define the mean reward as $\mathrm{R}(\mathcal{T}_T) = \frac{1}{T}\sum_{t=1}^{T} r_t$.[1] The transition $p : (\boldsymbol{s}_t, a_t) \rightsquigarrow \boldsymbol{s}_{t+1}$ can be deterministic (a function) or probabilistic (generative). The transition may either be the real system $p = p_{\mathrm{real}}$ or a system model $p = \hat{p}$. When the model $\hat{p}$ is probabilistic, besides the point prediction $\mathbb{E}\left\{\hat{p}(\boldsymbol{s}_{t+1}|(\boldsymbol{s}_t, a_t))\right\}$, it also provides information on the uncertainty of the prediction and/or to model the randomness of the system (Deisenroth & Rasmussen, 2011; Chua et al., 2018). Finally, in the description of the algorithms we index a trace $\mathcal{T} = \left((\boldsymbol{s}_1, \boldsymbol{a}_1), \ldots, (\boldsymbol{s}_T, \boldsymbol{a}_T)\right)$ as follows: for $t \in \{1, \ldots, T\}$, $\mathcal{T}_s[t] = \boldsymbol{s}_t$ and $\mathcal{T}_a[t] = \boldsymbol{a}_t$.

## 3.2 A NOTE ON TERMINOLOGY

By *model* we will consistently refer to the learned transition or system model $\hat{p}$ (never to any policy). *Rollout* is the procedure of obtaining a trace $\mathcal{T}$ from an initial state $\boldsymbol{s}_1$ by alternating a model or real system $p$ and a policy $\pi$ (Fig 2). We decided to rename what Silver et al. (2017) calls the prior policy to *guide* since prior clashes with Bayesian terminology (as, e.g., Grill et al. (2020); Hamrick et al. (2021) also note), and guide expresses well that the role of this policy is to guide the search at decision time. Sometimes the guide is also called the *reactive* policy (Sun et al., 2018) since it is typically an explicit function or conditional distribution $\xi : \boldsymbol{s} \rightsquigarrow a$ that can be executed or drawn from rapidly. We will call the (often implicit) policy $\pi : \boldsymbol{s} \rightsquigarrow a$ resulting from the guided plan/search the *actor* (sometimes also called the *non-reactive* policy since it takes time to simulate from the model, before each action). *Planning* generally refers to the use of a model $\hat{p}$ to generate imaginary plans and in that sense, planning is part of training the guide. However, in the rest of the paper we will use the term *planning* to refer to the guided search procedure that results in acting on the real system (this is sometimes called *decision-time* planning). We will explicitly use the term *background* planning to refer to the planning used at training time as it is done in Sutton & Barto (2018) and Hamrick et al. (2021).

## 3.3 EXPERIMENTAL SETUP: THE ITERATED BATCH MBRL

[2]For rigorously studying and comparing algorithms and algorithmic ingredients, we need to fix not only the simulation environment but also the experimental setup. We parameterize the iterated batch RL loop (the pseudocode in Fig 2 is the formal definition) by four parameters:

- the number of episodes $N$,
- the number of system access steps $T$ per episode,
- the planning horizon $L$, and
- the number of generated rollouts $n$ at each planning step.

$N$ and $T$ are usually set by hard organizational constraints (number $N$ and length $T$ of live tests) that are part of the experimental setup. Our main goal is to measure the performance of our algorithms at a given (and challengingly small) number of system access steps $N \times T$ for a given planning budget ($n$ and $L$) determined by the (physical) time between two steps and the available computational resources.

In benchmark studies, we argue that fixing $N$, $T$, $n$, and $L$ is important for making the problem well defined (taking some of the usual algorithmic choices out of the input of the optimizer), affording meaningful comparison across papers and steady progress of algorithms. As in all benchmark designs, the goal is to make the problem challenging but not unsolvable. That said, we are aware that these choices may change the task and the research priorities implicitly but significantly (for example, a longer horizon $L$ will be more challenging for the model but may make the planning easier although more expensive), so it would make sense to carefully design several settings (quadruples $N - T - n - L$) on the same environment.

---

[1]We use the *mean* reward (as opposed to the *total* reward, a.k.a return), since it is invariant to episode length and its unit is more meaningful to systems engineers.

[2]First a practical note: the pseudocode in Fig 2 and the subroutines in Figs 3-5 contain our formal definition. They are ordered top-down, but can also be read in reverse order according to the reader's preference.

$\text{ROLLOUT}(\pi, p, \boldsymbol{s}_1, T)$:
1 $\mathcal{T} \leftarrow \{\}$
2 **for** $t \leftarrow 1$ **to** $T$:                                         ▷ *for $T$ steps*
3    $a_t \curvearrowleft \pi(\boldsymbol{s}_t)$                       ▷ *draw action from policy*
4    $\mathcal{T} \leftarrow \mathcal{T} \cup (\boldsymbol{s}_t, a_t)$   ▷ *update trace*
5    $\boldsymbol{s}_{t+1} \curvearrowleft p(\boldsymbol{s}_t, a_t)$     ▷ *draw next state*
6 **return** $\mathcal{T}$

$\text{ITERATEDMBRL}(p_{\text{real}}, \mathcal{S}_0, \pi^{(0)}, N, T, L, n)$:
1 $\boldsymbol{s}_1 \curvearrowleft \mathcal{S}_0$                               ▷ *draw initial state*
2 $\mathcal{T}^{(1)} \leftarrow \text{ROLLOUT}\left(\pi^{(0)}, p_{\text{real}}, \boldsymbol{s}_1, T\right)$   ▷ *random trace*
3 **for** $\tau \leftarrow 1$ **to** $N$:                                      ▷ *for $N$ episodes*
4    $\hat{p}^{(\tau)} \leftarrow \text{LEARN}\left(\cup_{\tau'=1}^{\tau} \mathcal{T}^{(\tau')}\right)$   ▷ *learn system model*
5    $\pi^{(\tau)} \leftarrow \text{ACTOR}\left(\pi^{(0)}, \pi^{(\tau-1)}, \hat{p}^{(\tau)}, \cup_{\tau'=1}^{\tau} \mathcal{T}^{(\tau)}, L, n\right)$
6    $\boldsymbol{s}_1 \curvearrowleft \mathcal{S}_0$              ▷ *draw initial state*
7    $\mathcal{T}^{(\tau+1)} \leftarrow \text{ROLLOUT}\left(\pi^{(\tau)}, p_{\text{real}}, \boldsymbol{s}_1, T\right)$   ▷ *new trace*
8 **return** $\cup_{\tau=1}^{N} \mathcal{T}^{(\tau)}$

Figure 2: **The iterated batch MBRL loop.** $p_{\text{real}} : (\boldsymbol{s}_t, a_t) \rightsquigarrow \boldsymbol{s}_{t+1}$ is the real system (so Line 7 is what dominates the cost) and $p : (\boldsymbol{s}_t, a_t) \rightsquigarrow \boldsymbol{s}_{t+1}$ is the transition in ROLLOUT that can be either the real system $p_{\text{real}}$ or the system model $\hat{p}$. $\mathcal{S}_0$ is the distribution of the initial state of the real system. $\pi^{(0)} : \boldsymbol{s}_t \rightsquigarrow a_t$ is an initial (typically random) policy and in ROLLOUT $\pi : \boldsymbol{s}_t \rightsquigarrow a_t$ is any policy. $N$ is the number of episodes; $T$ is the length of the episodes; $L$ is the planning horizon and $n$ is the number of planning trajectories used by the actor policies $\pi^{(\tau)}$. $\tau = 1, \ldots, N$ is the episode index whereas $t = 1, \ldots, T$ is the system (or model) access step index. LEARN is a supervised learning (probabilistic or deterministic time-series forecasting) algorithm applied to the collected traces and ACTOR is a wrapper of the various techniques that we experiment with in this paper (Fig 4). An ACTOR typically updates $\pi^{(\tau-1)}$ using the model $\hat{p}^{(\tau)}$ in a background-planning loop, but it can also access the initial policy $\pi^{(0)}$ and the trace $\cup_{\tau'=1}^{\tau} \mathcal{T}^{(\tau')}$ collected on $p_{\text{real}}$ up to episode $\tau$.

$\text{HEATINGEXPLORE}\left(\pi^{(0)}, \xi, n\right)[\boldsymbol{s}]$:
1 **for** $i \leftarrow 1$ **to** $n$:
2    $\rho_i(\boldsymbol{a}|\boldsymbol{s}) \sim \begin{cases} \dfrac{\xi(\boldsymbol{a}|\boldsymbol{s})^{1/T_i}}{\sum_{\boldsymbol{a}'} \xi(\boldsymbol{a}'|\boldsymbol{s})^{1/T_i}} & \text{if } \boldsymbol{a} \text{ is discrete} \\[2ex] \mathcal{N}\left(\mathbb{E}\{\xi(.|s)\}, T_i\right) & \text{if } \boldsymbol{a} \text{ is continuous.} \end{cases}$
3 **return** $[\rho_i]_{i=1}^{n}$

Figure 3: **Heating exploration strategy.** HEATINGEXPLORE heats the guide action distribution $\xi(\boldsymbol{a}|\boldsymbol{s})$ to $n$ different temperatures. The temperatures $[T_i]_{i=1}^{n}$ are hyperparameters.

Our main operational cost is system access step so we are looking for any-time algorithms that achieve the best possible performance at any episode $\tau$ for a given decision-time planning budget ($n$ and $L$). Hence, in the MBRL iteration (Fig 2), we use the same traces $\mathcal{T}^{(\tau)}$, rolled out in each iteration (Line 7), to i) update the model $\hat{p}$ and the actor policy (Line 4 and 5) and ii) to measure the performance of the techniques (Section 3.5).

### 3.4 MODEL-BASED ACTOR POLICIES: GUIDE AND EXPLORE

Our main contribution is a Dyna-style GUIDE&EXPLORE strategy (Fig 4). This strategy consists in learning a guide policy $\xi$ for the decision-time planner (TRAJOPT in Fig 5) using a model-free RL technique on the model $\hat{p}$ and on the traces collected on the real system $\mathcal{T}$. It is known that the guide $\xi$, executed as an actor $\pi = \xi$ on the real system, does not usually give the best performance (we also confirm it in Section 4), partly because $\xi$ overfits the model (Fig 5 in Kurutach et al. (2018); Grill et al. (2020)), partly because the goal of $\pi$ is not only to exploit the traces $\mathcal{T} = \cup_{\tau=1}^{N} \mathcal{T}^{(\tau)}$ collected so far and the model $\hat{p} = \text{LEARN}(\mathcal{T})$, but also to collect data to self-improve $\hat{p}$ and $\xi/\pi$ in the next episode $\tau$. This second reason is particular in our *iterated* batch setup: contrary to *pure* batch

$\text{RSACTOR}\big(\pi^{(0)}, \pi^{\text{prev}}, p, \mathcal{T}, L, n\big)$:
1 **return** $\text{TRAJOPT}\big([\pi^{(0)}]_{i=1}^n, p, L, n\big)$      ▷ *planning*

$\text{GUIDE\&EXPLOREACTOR}\big(\pi^{(0)}, \pi^{\text{prev}}, p, \mathcal{T}, L, n\big)$:
1 $\xi \leftarrow \text{MFRL}\big(\pi^{\text{prev}}, p, \mathcal{T}\big)$      ▷ *model free guide policy*
2 $[\rho_i]_{i=1}^n \leftarrow \text{HEATINGEXPLORE}\big(\pi^{(0)}, \xi, n\big)$      ▷ *explorer policies*
3 **return** $\text{TRAJOPT}\big([\rho_i]_{i=1}^n, p, L, n\big)$      ▷ *planning*

Figure 4: **Model-based ACTORS (policies executed on the real system).** RSACTOR is a classical random shooting planner that uses the random policy $\pi^{(0)}$ for all rollouts. GUIDE&EXPLOREACTOR first learns a Dyna-style *guide* policy $\xi$ on the transition $p$ (more precisely, updates the previous guide contained in $\pi^{\text{prev}}$). It can also use the traces $\mathcal{T}$ collected on the real system. It then "decorates" the guide by (possibly $n$ different) exploration strategies, and runs these reactive guide&explore policies $[\rho_i]_{i=1}^n$ in the TRAJOPT planner in Fig 5.

$\text{TOTALREWARD}\big(\mathcal{T}\big)$:
1 **return** $T \times \text{R}\big(\mathcal{T}\big)$      ▷ *total reward (a.k.a return)*

$\text{BOOTSTRAP}\big(V, \alpha\big)\big(\mathcal{T}\big)$:
1 **return** $T \times \text{R}\big(\mathcal{T}\big) + \alpha V\big(\mathcal{T}_s[L]\big)$      ▷ *total reward + value of last state*

$\text{TRAJOPT}\big([\rho_i]_{i=1}^n, p, L, n\big)[\boldsymbol{s}]$:
1 **for** $i \leftarrow 1$ **to** $n$:
2    $\mathcal{T}^{(i)} \leftarrow \text{ROLLOUT}\big(\rho_i, p, \boldsymbol{s}, L\big)$      ▷ *ith roll-out trace*
3    $V^{(i)} \leftarrow \text{VALUE}\big(\mathcal{T}^{(i)}\big)$      ▷ *total reward of $\mathcal{T}^{(i)}$*
4 $i^* \leftarrow \text{argmax}_i V^{(i)}$      ▷ *index of the best trace*
5 **return** $\boldsymbol{a}_1^* = \mathcal{T}_a^{(i^*)}[1]$      ▷ *first action of the best trace*

Figure 5: **VALUE estimates on rollout traces and TRAJOPT: trajectory optimization using random shooting with a *set* of policies.** TOTALREWARD and BOOSTRAP are two ways to evaluate the value of a rollout trace. The latter adds the value of the last state to the total reward, according to a value estimate $V : \boldsymbol{s} \to \mathbb{R}^+$, weighted by a hyperparameter $\alpha$. They are called in Line 3 of TRAJOPT which is a random shooting planner that accepts $n$ different shooting policies $[\rho_i]_{i=1}^n$ for the $n$ rollouts used in the search. As usual, it returns the first action $\boldsymbol{a}_1^* = \mathcal{T}_a^{(i^*)}[1]$ of the best trace $\mathcal{T}^{(i^*)} = \big((\boldsymbol{s}_1^*, \boldsymbol{a}_1^*), \dots, (\boldsymbol{s}_T^*, \boldsymbol{a}_T^*)\big)$. Its parameters are the shooting policies $[\rho_i]_{i=1}^n$, the transition $p$, and the number $n$ and length $L$ of rollouts, but to properly define it, we also need the state $\boldsymbol{s}$ which we plan from, so we use a double argument list $()[]$. We recall here that for a trace $\mathcal{T} = \big((\boldsymbol{s}_1, \boldsymbol{a}_1), \dots, (\boldsymbol{s}_T, \boldsymbol{a}_T)\big)$ and for $t \in \{1, \dots, T\}$, $\mathcal{T}_s[t] = \boldsymbol{s}_t$ and $\mathcal{T}_a[t] = \boldsymbol{a}_t$.

RL, exploration here is crucial. When the guide $\xi$ is probabilistic, we explore implicitly because of the sampling step (Line 3 in ROLLOUT, Fig 1), and partly also because of the draw from the imperfect and possibly stochastic model (Line 5). Nevertheless, we found that it helps if we control exploration explicitly. To show this, we experiment with a HEATINGEXPLORE strategy which consists in modulating the temperature of the guide distribution $\xi(\boldsymbol{a}|\boldsymbol{s})$ (Fig 4). The novelty of our approach is that, instead of constant randomness in the exploration, we use a *set* of temperatures $[T_i]_{i=1}^n$ to further diversify the search and let the planner explore promising regions far from the distribution of trajectories where the guide may have falsely converged. Finally, similarly to Lowrey et al. (2019); Argenson & Dulac-Arnold (2021), we found that bootstrapping the planning with the learned value function at the end of each rollout trace (BOOTSTRAP in Fig 5) can be helpful for optimizing the performance with a short horizon.

## 3.5 METRICS

We use two rigorously defined and measured metrics (Kégl et al., 2021) to assess the performance of the different algorithmic combinations. MAR measures the asymptotic performance after the

learning has converged, and MRCP measures the convergence pace. Both can be averaged over seeds, and MAR is also an average over episodes, so we can detect statistically significant differences even when they are tiny, leading to a proper support for experimental development.

**MEAN ASYMPTOTIC REWARD (MAR).** Our measure of asymptotic performance, the mean asymptotic reward, is the mean reward $\mathrm{MR}(\tau) = \mathrm{R}\left(\mathcal{T}_T^{(\tau)}\right)$ in the second half of the episodes (after convergence; we set $N$ in such a way that the algorithms converge after less than $N/2$ episodes) $\mathrm{MAR} = \frac{2}{N} \sum_{\tau=N/2}^{N} \mathrm{MR}(\tau)$.

**MEAN REWARD CONVERGENCE PACE (MRCP($\bar{r}$)).** To assess the speed of convergence, we define the mean reward convergence pace $\mathrm{MRCP}(\bar{r})$ as the number of steps needed to achieve mean reward $\bar{r}$, smoothed over a window of size 5: $\mathrm{MRCP}(\bar{r}) = T \times \mathrm{argmin}_\tau \left( \frac{1}{5} \sum_{\tau'=\tau-2}^{\tau+2} \mathrm{MR}(\tau) > \bar{r} \right)$. The unit of $\mathrm{MRCP}(\bar{r})$ is system access steps, not episodes, first to make it invariant to episode length, and second because in micro-data RL the unit of cost is a system access step. For Acrobot, we use $\bar{r} = 1.8$ in our experiments, which is roughly 70% of the best achievable mean reward.

## 4 EXPERIMENTS

### 4.1 ACROBOT

Acrobot is an underactuated double pendulum with four observables $\boldsymbol{s}_t = [\theta_1, \theta_2, \dot{\theta}_1, \dot{\theta}_2]$ which are usually augmented to six by taking sine and cosine of the angles (Brockman et al., 2016); $\theta_1$ is the angle to the vertical axis of the upper link; $\theta_2$ is the angle of the lower link relative to the upper link, both being clipped to $[-\pi, \pi]$; $\dot{\theta}_1$ and $\dot{\theta}_2$ are the corresponding angular momenta. For the starting position $\boldsymbol{s}_1$ of each episode, all four state variables are sampled uniformly from an approximately hanging and stationary position $s_1^j \in [-0.1, 0.1]$. The action is a discrete torque on the lower link $a \in \{-1, 0, 1\}$. The reward is the height of the tip of the lower link over the hanging position $r(\boldsymbol{s}) = 2 - \cos\theta_1 - \cos(\theta_1 + \theta_2) \in [0, 4]$.[3]

Acrobot is a small but relatively difficult and fascinating system, so it is an ideal benchmark for continuous-reward engineering systems. Similarly to Kégl et al. (2021), we set the number of episodes to $N = 100$, the number of steps per episode to $T = 200$, the number of planning rollouts to $n = 100$, and the horizon to $L = 10$. With these settings, we can identify four distinctively different regimes (see the attached videos): i) the random uniform policy $\pi^{(0)}$ achieves $\mathrm{MAR} \approx 0.1 - 0.2$ (Acrobot keeps approximately hanging), ii) reasonable models with random shooting or pure Dyna-style controllers achieve $\mathrm{MAR} \approx 1.4 - 1.6$ (Acrobot gains energy but moves its limb quite uncontrollably), iii) random shooting $n = 100, L = 10$ with good models such as PETS (Chua et al., 2018; Wang et al., 2019) or DARMDN (Kégl et al., 2021) keep the limb up and manage to have its tip above horizon on average $\mathrm{MAR} \approx 2.0 - 2.1$ (previous state of the art), and iv) in our experiments we could achieve a quasi perfect policy (Acrobot moves up like a gymnast and stays balanced at the top) $\mathrm{MAR} \approx 2.7 - 2.8$ using random shooting with $n = 100\mathrm{K}, L = 20$ on the *real system*, giving us a target and a possibly large margin of improvement.

Acrobot is an ideal benchmark for making our point for the two following reasons. First, it turned out to be a quite difficult system for pure model-free baselines and associated Dyna-style algorithms (we achieve a $\mathrm{MAR} \approx 2.1$ with a DQN, see Table 1, and Wang et al. (2019) report a $\mathrm{MAR} \approx 1.6 - 1.7$ for other Dyna-style algorithms). Second, decision-time planning can achieve a quasi-perfect policy ($\mathrm{MAR} \approx 2.7 - 2.8$) but doing so while being data efficient and with a limited planning budget appears to be challenging.

### 4.2 CARTPOLE SWING-UP

Cartpole swing-up from the DeepMind Control Suite (Tunyasuvunakool et al., 2020) is an underactuated pendulum attached by a frictionless pivot to a cart that moves freely along a horizontal line. Observations include the cart position, cosine and sine of the pendulum angle, and their time derivatives $\boldsymbol{s}_t = [x, \dot{x}, \cos\theta, \sin\theta, \dot{\theta}]$. The cart is initialized at a position $x$ and a velocity close to

---

[3]We chose this rather than the sparse variable-episode-length version $r(\boldsymbol{s}) = \mathbb{I}\{2 - \cos\theta_1 - \cos(\theta_1 + \theta_2) > 3\}$ (Sutton, 1996) since it corresponds better to the continuous aspect of engineering systems.

0, and an angle close to $\theta = \pi$ (hanging position). The goal is to swing-up the pendulum and stabilize it upright by applying a continuous horizontal force $a_t \in [-1, 1]$ to the cart at each timestep $t$. The reward in $[0, 1]$ is obtained by the multiplication of four reward components: one depending on the height of the pendulum (in $[1/2, 1]$), one on the cart position (in $[1/2, 1]$), one on its velocity (in $[1/2, 1]$) and one on the amplitude of the force (in $[4/5, 1]$). The maximum reward is obtained when the pendulum is centered ($x = 0$), upright with no velocity and an applied force of 0. This task has been widely used by the literature as a standard benchmark system for nonlinear control and complex action space due to its potential generalization to different domains (Boubaker, 2012; Nagendra et al., 2017). We set the number of episodes to $N = 35$, the number of steps per episode to $T = 1000$, the number of planning rollouts to $n = 500$, and the horizon to $L = 20$. A mean reward of 0.8 corresponds to a pole that succeeds at standing upright and being stable.

We chose the Cartpole system as vanilla MPC agents (RS or CEM) require a long planning horizon (at least $L = 100$) to succeed at swinging up the pendulum and stabilize it upright. It also illustrates how our approach extends to the continuous action setting which increases the complexity of the optimization search space and requires sensitive controllers.

### 4.3 MODELS, GUIDES, AND ACTORS

Following Kégl et al. (2021), we tried different system models (Fig 2/Line 4) from the family of Deep (Autoregressive) Mixture Density Networks (D(A)RMDN) and selected the ones giving the best results on the Acrobot and Cartpole swing-up systems.

In principle, any MFRL technique providing a value function (for bootstrapping) and a policy can be used as a guide $\xi$ when applying ITERATEDMBRL (Fig 4/Line 1). We argue though that an off-policy algorithm is better suited here since it can leverage all the (off-policy) data coming from interaction with the real system. In particular, those traces are generated using planning and represent a stronger learning signal for the agent (planning during training vs. at test time; Hamrick et al. (2021)). Thus, we experimented with Deep Q Networks (DQN; Mnih et al. (2015)) for the discrete action Acrobot system and Soft Actor-Critic (SAC) (Haarnoja et al., 2018) for the continuous action Cartpole swing-up task. For SAC on Cartpole, following Janner et al. (2019), short rollouts starting from real observations are performed on the model to sample transitions which are then placed in an experience replay buffer, along with the real transitions observed during the rollouts (Fig 2/Line 7). The SAC is then updated by sampling batches from this buffer. Details on the implementation and hyperparameters of the DQN and SAC agents are given in Appendix A.

For actors (Fig 2/Line 5, Fig 4), we start from the simple model-free guide (DQN or SAC) which is trained with data generated by the model and is interacting with the system without planning. We refer to these agents as MBPO(DQN) and MBPO(SAC). Adding ($n$, $L$) to the name of the agent means that we use the agent to guide a planning of $n$ rollouts with horizon $L$.

Appendix A contains detailed information on the various algorithmic choices.

### 4.4 RESULTS

Table 1 and Fig 6 compare the results obtained with GUIDE&EXPLORE, a vanilla RSACTOR agent using the same budget as the one we consider for GUIDE&EXPLORE, the pure model free guides trained on the real system and their Dyna-style version (no planning, MBPO). We see that the GUIDE&EXPLORE algorithm gives the best performance. On Acrobot it is almost matching the costly RSACTOR($n = 100K, L = 20$) that we include as a target even though it would not be officially accepted in our benchmark as we restrict $n$ to 100 and $L$ to 10. On Cartpole it reaches the performance reported in Lee et al. (2020) and approaches the one from (Springenberg et al., 2020) which are, to the best of our knowledge, the state of the art on Cartpole. We note that Springenberg et al. (2020) reports the median performance whereas we report the mean performance as Lee et al. (2020).

We ran an ablation study on Acrobot showing that *all* ingredients add to the performance. Although MBPO(DQN) performs reasonably well and is comparable to RSACTOR($n = 100, L = 10$) it fails to achieve the performance of RSACTOR($n = 100K, L = 20$). Fig 6 also shows that adding a heating explorer to the guide significantly improves the performance of RSACTOR($n = 100, L = 10$) and MBPO(DQN). We found that allowing the planner to choose the right amount of exploration (the right temperature) is a robust and safe approach (see Appendix B and Fig 8 for more results with a

Table 1: Agent evaluation results. MAR is the mean asymptotic reward showing the asymptotic performance of the agent and MRCP(1.8) is the mean reward convergence pace showing the sample-efficiency (the number of system access steps required to achieve a mean reward of 1.8). The model-free agents are trained on the real system and the corresponding MAR shows the asymptotic performance obtained after convergence. ↓ and ↑ mean lower and higher the better, respectively.

| Acrobot | MAR ↑ | MRCP(1.8) ↓ |
|---|---|---|
| MBPO(DQN) | 2.113±0.02 | 4140 ±720 |
| RSACTOR($n = 100, L = 10$) | 2.075±0.01 | 2620 ±320 |
| **DQN($n = 100, L = 10$)-GUIDE&EXPLORE** | **2.404**±0.017 | 1800 ±290 |
| RSACTOR($n = 100K, L = 20$) | 2.474±0.022 | 2280 ±580 |
| Model free DQN | 2.16 ±0.018 | 1.58M±0.20M |

| Cartpole | MAR ↑ | MRCP(0.5) ↓ |
|---|---|---|
| MBPO(SAC) | 0.566±0.055 | 17750±7430 |
| RSACTOR($n = 500, L = 20$) | 0.304±0.01 | inf ±inf |
| **SAC($n = 500, L = 20$)-GUIDE&EXPLORE** | **0.732**±0.021 | 10500±5020 |
| Model free SAC | 0.781±0.05 | 72250±11000 |

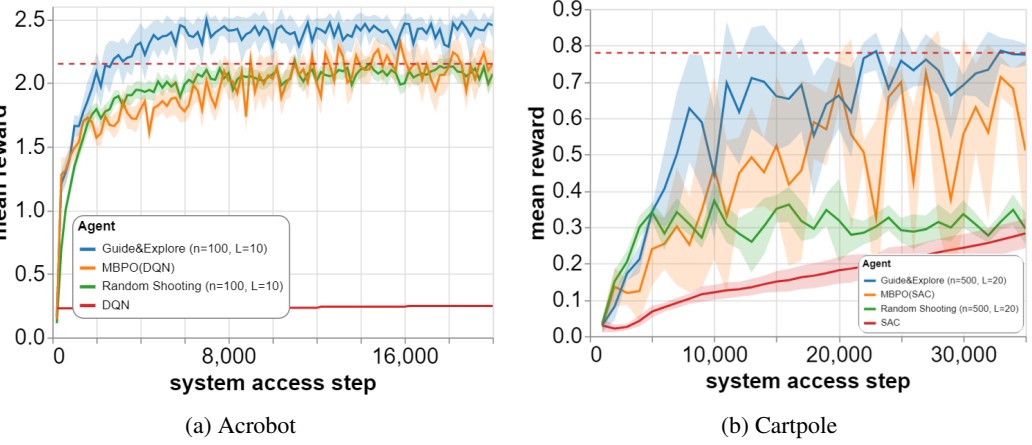

| (a) Acrobot | (b) Cartpole |
|---|---|

Figure 6: Learning curves obtained with different agents. Mean reward curves are averaged across at least four seeds. Areas with lighter colors show the 90% confidence intervals and dashed lines represent the score of the best converged model-free algorithms. (a) Mean reward is between 0 (hanging) and 4 (standing up). Episode length is $T = 200$, number of epochs is $N = 100$ with one episode per epoch. (b) Mean reward is between 0 and 1. Episode length is $T = 1000$, number of epochs is $N = 35$ with one episode per epoch.

suboptimal DQN guide). The final improvement was attained by adding value bootstrapping to the heated explorer/planner.

## 5 CONCLUSION

In this paper we show that an offline Dyna-style approach can be successfully applied on benchmark systems where previously Dyna-style algorithms were failing. Our empirical results exhibit the importance of guiding the decision-time planning with the correct amount of exploration to achieve the best performance under a planning budget constraint. More precisely, our decision-time planner explores a varied range of trajectories around the guide's prior distribution and bootstraps with a value function estimate to further improve the performance.

This combination leads to achieving state-of-the-art performance while respecting reasonable resource constraints. Future work includes modelling the uncertainties of the value estimates so as to use them for better exploration.

## REPRODUCIBILITY STATEMENT

In order to ensure reproducibility we will release the code at `<URL hidden for review>`, once the paper has been accepted. We also provide details on the hyperparameter optimization of the agents and models as well as the best ones in the Appendix.

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

## A  IMPLEMENTATION DETAILS

### A.1  CODE AND DEPENDENCIES

Our code will be made publicly available after publication to ease the reproducibility of all our results. We use Pytorch (Paszke et al., 2019) to build and train the neural network system models and policies. To run the ITERATEDMBRL experiments we use the `rl_simulator` (https://github.com/ramp-kits/rl_simulator) python library developed by Kégl et al. (2021) which relies on Open AI Gym (Brockman et al., 2016) for the Acrobot dynamics and `dm_control` (Tunyasuvunakool et al., 2020) for the Cartpole swing-up task. For the DQN and SAC agents we rely on the StableBaselines3 implementations (Raffin et al., 2019).

### A.2  MODELS AND AGENTS

It is known that carefully tuning hyperparameters of deep reinforcement learning algorithms is crucial for success and fair comparisons (Henderson et al., 2018; Zhang et al., 2021). To reduce the computational cost and consider a reasonable search space the models and the agents were optimized independently. For the Acrobot system models we use the same hyperparameters as the ones used in Kégl et al. (2021). Please refer to Appendix D in Kégl et al. (2021) for a complete description of the hyperparameter search and the selected hyperparameters. We decided to use $DMDN(1)_{det}$ trained on the first 2000 and last 3000 collected samples as it lead to a similar performance with a limited training time. For Cartpole we use a DARMDN model with 1 hidden layer and 128 neurons trained on the full dataset. These models are trained by minimizing the negative log-likelihood. The 'det' suffix means that the model is sampled from deterministically, returning the mean of the predicted distribution. The reader can also refer to Kégl et al. (2021) for a complete description of these models.

For the DQN we optimized its hyperparameters with a random search of 1000 trials and a parallel training on 10 copies of the real system for 10 million steps (Table 2). We then selected the DQN with the best mean reward (Table 3). When training DQN on the system model, we iteratively update it with 100 000 steps at each episode using the most recent system model. We do not use short rollouts as this was not necessary. When bootstrapping with the value function we used a discount factor of 0.95 as it lead to the best performance.

For the SAC agent we performed a similar hyperparameter optimization with a random search of 1000 trials and a parallel training on 10 copies of the real system for 1 million steps (Table 4). The best SAC parameters are given in Table 5. When training SAC on the system model, we train it from scratch at each episode for 250 000 steps and perform short rollouts of length 100.

Table 2: Random search parameters for the model free DQN

| Parameter | Values |
|---|---|
| Discount factor | [0.9, 0.99, 1] |
| Polyak update parameter | [0.1, 0.5, 1] |
| Learning rate | [0.0001, 0.001] |
| Final training epsilon value | [0.01, 0.1] |
| Exploration fraction | [0.1, 0.5] |
| Buffer size | [10000, 100000] |
| Update frequency of the network | [2, 4, 10] |
| Batch size | [64, 128] |
| Update frequency of the target network | [500, 1000] |
| Network architecture | [MLP(128, 128), MLP(256, 256] |
| Gradient steps | [1, 2, 5, 10] |

### A.3  EXPLORATION

The multi-temperature HEATINGEXPLORE strategy depends on whether actions are discrete or continuous:

Table 3: Best parameters for the model free DQN

| Parameter | Values |
|---|---|
| Discount factor | 0.99 |
| Polyak update parameter | 0.5 |
| Learning rate | 0.001 |
| Final training epsilon value | 0.01 |
| Exploration fraction | 0.1 |
| Buffer size | 10000 |
| Update frequency of the network | 4 |
| Batch size | 128 |
| Update frequency of the target network | 1000 |
| Network architecture | MLP(256, 256) |
| Gradient steps | 10 |

Table 4: Random search parameters for the model free SAC

| Parameter | Values |
|---|---|
| Discount factor | [0.9, 0.95, 0.99, 1] |
| Polyak update parameter | [0.1, 0.5, 1] |
| Learning rate | [0.0001, 0.0005, 0.001] |
| Buffer size | [10000, 100000, 1000000] |
| Update frequency of the network | [4, 10, 20] |
| Batch size | [64, 128] |
| Update frequency of the target network | [100, 1000, 10000] |
| Network architecture | [MLP(64, 64), MLP(128, 128), MLP(256, 256] |
| Gradient steps | [1, 5, 10, 20] |

Table 5: Best parameters for the model free SAC

| Parameter | Values |
|---|---|
| Discount factor | 1.0 |
| Polyak update parameter | 0.1 |
| Learning rate | 0.0005 |
| Buffer size | 1000000 |
| Update frequency of the network | 10 |
| Batch size | 128 |
| Update frequency of the target network | 1000 |
| Network architecture | MLP(256, 256) |
| Gradient steps | 20 |

- For environments with discrete actions, we learn a $Q$ function and first normalize the $Q$ values by their maximum value, $\tilde{Q}(s,a) = Q(s,a)/\max_{a'} Q(s,a')$, before applying a softmax:

$$\rho_i(a|s_t) = \frac{e^{\tilde{Q}(s_t,a)/T_i}}{\sum_{a'} e^{\tilde{Q}(s_t,a')/T_i}}$$

- For continuous actions environments, we learn a Gaussian policy $\xi(.|s) \sim \mathcal{N}\big(\mu(s), \sigma(s)\big)$ and define the explorer policies as:

$$\rho_i(.|s_t) \sim \mathcal{N}\big(\mu(s), T_i\big)$$

$\{T_i, 1 \leq i \leq n\}$ is an increasing sequence of temperatures. A large temperature gives a uniform distribution, whereas a low temperature corresponds to taking $\operatorname{argmax}_a Q(s_t, a)$ or $\mu(s)$. Different shapes of sequences were tried (linear, logarithmic, exponential, polynomial, logistic), and best

performance was obtained with a logistic schedule (with a linear end). The exact values will be provided in the code.

For Acrobot and the DQN guide we also played with a multi-$\varepsilon$ exploration strategy based on EPSGREEDYEXPLORE (Fig 7) where we use one $\varepsilon$ value for each of the $n = 100$ rollouts: $\{0.001, 0.01, 0.02, \ldots, 0.99\}$. Refer to Appendix B for experiments with a suboptimal DQN guide using this exploration strategy.

---

EPSGREEDYEXPLORE$\big(\pi^{(0)}, \xi, n\big)[\boldsymbol{s}]$:
1 **for** $i \leftarrow 1$ **to** $n$:
2 $\quad \rho_i(\boldsymbol{a}|\boldsymbol{s}) = \begin{cases} \operatorname{argmax}_a \xi(\boldsymbol{a}|\boldsymbol{s}) & \text{with probability } 1 - \varepsilon_i \\ \pi^{(0)}(\boldsymbol{a}) & \text{with probability } \varepsilon_i. \end{cases}$
3 **return** $[\rho_i]_{i=1}^n$

---

Figure 7: **Exploration strategies.** EPSGREEDYEXPLORE changes the action to a random action $\pi^{(0)} \rightsquigarrow \boldsymbol{a}$ with different probabilities. The probabilities $[\varepsilon_i]_{i=1}^n$ are hyperparameters.

## B IMPORTANCE OF THE EXPLORATION: STUDY WITH A SUBOPTIMAL DQN GUIDE ON ACROBOT

We ran an ablation study with a suboptimal DQN guide on Acrobot (achieving an asymptotic performance of 1.6 on the real system) and a multi-$\varepsilon$ greedy explorer (EPSGREEDYEXPLORE) to claim the importance of the explorer. EPSGREEDYEXPLORE makes it easy to control and interpret the degree of exploration through the $\varepsilon$ parameter.

We consider the following agents. We start from the simple DQN guide which is interacting with the system without planning. Adding $(n, L)$ to the name of the agent means that we use the agent to guide a planning of $n$ rollouts with horizon $L$. It is important to note here that planning without exploration using the greedy guide is, in our case, equivalent to no planning since both the model $\hat{p}$ and the guides $\xi$ are deterministic. DQN-EPSGREEDYEXPLORE refers to the additional use of the associated exploration strategy (Fig 7). When a fixed $\varepsilon$ is used for the exploration strategy, we add it as an explicit parameter, e.g., EPSGREEDYEXPLORE$(\varepsilon)$. No parameters means that a different $\varepsilon$ or temperature is used for each of the $n$ rollouts. Setting $\varepsilon$ to 0 corresponds to no exploration and is equivalent to using the guide greedily without planning ($n = 1$ and $L = 1$) as our model is used deterministically when sampled from. Setting $\varepsilon$ to 1 corresponds to full exploration and is equivalent to the purely random RSACTOR($n = 100, L = 10$).

Table 6 reports the results obtained by DQN alone, DQN with planning and fixed $\varepsilon$ values (DQN($n = 100, L = 10$)-EPSGREEDYEXPLORE$(\varepsilon)$ for $\varepsilon \in \{0.0001, 0.01, 0.05, 0.1, 0.2, 0.4, 0.8, 0.99, 0.9999\}$). The closer $\varepsilon$ is to 0 the closer the performance is to DQN, and the closer $\varepsilon$ is to 1 the closer the performance is to RSACTOR($n = 100, L = 10$). With a well-chosen $\varepsilon$ between these two extremes, say $\varepsilon = 0.4$, we obtain a better performance than either extremes. We can thus claim that planning is required with a correct amount of exploration. Our EPSGREEDYEXPLORE exploration strategy, used with multiple $\varepsilon$ values allows for the automatic and dynamic selection of the good amount of exploration. It would indeed be possible that using a fixed $\varepsilon$ value would not give the best performance as different values would be required at different epochs or different steps of an episode. We illustrate this by plotting the selected epsilon vs the episode step for 3 different epochs (Fig 8). Even though the guide is suboptimal the exploration scheme can make the agent benefit from the guide where it is good and discard it where it is bad.

## C RSACTOR PERFORMANCE ON THE REAL ACROBOT AND CARTPOLE SYSTEMS

### C.1 ACROBOT

We present the results one can obtain on the real system with an RSACTOR and different values of the planning horizon $L$ and the number of generated rollouts $n$ in Fig 9. For the considered planning

Table 6: Importance of the explorer. MAR is the Mean Asymptotic Reward showing the asymptotic performance of the agent and MRCP(1.8) is the Mean Reward Convergence Space showing the sample-efficiency performance as the number of system access steps required to achieve a reward of 1.8. ↓ and ↑ mean lower and higher the better, respectively. The ± values are 90% Gaussian confidence intervals.

| Agent | MAR ↑ | MRCP(1.8) ↓ |
|---|---|---|
| DQN | 1.442±0.014 | inf  ±inf |
| DQN($n = 100$, $L = 10$)-EPSGREEDYEXPLORE($\varepsilon = 0.0001$) | 1.475±0.062 | inf  ±inf |
| DQN($n = 100$, $L = 10$)-EPSGREEDYEXPLORE($\varepsilon = 0.01$) | 1.664±0.032 | inf  ±inf |
| DQN($n = 100$, $L = 10$)-EPSGREEDYEXPLORE($\varepsilon = 0.05$) | 1.932±0.012 | 3540±520 |
| DQN($n = 100$, $L = 10$)-EPSGREEDYEXPLORE($\varepsilon = 0.1$) | 2.009±0.031 | 2400±− |
| RSACTOR($n = 100$, $L = 10$) | 2.075±0.01 | 2620±320 |
| DQN($n = 100$, $L = 10$)-EPSGREEDYEXPLORE($\varepsilon = 0.9999$) | 2.107±0.042 | 2000±− |
| DQN($n = 100$, $L = 10$)-EPSGREEDYEXPLORE($\varepsilon = 0.99$) | 2.118±0.046 | 2400±− |
| DQN($n = 100$, $L = 10$)-EPSGREEDYEXPLORE($\varepsilon = 0.2$) | 2.151±0.034 | 2400±− |
| DQN($n = 100$, $L = 10$)-EPSGREEDYEXPLORE($\varepsilon = 0.8$) | 2.196±0.037 | 2000±− |
| DQN($n = 100$, $L = 10$)-EPSGREEDYEXPLORE($\varepsilon = 0.4$) | 2.204±0.01 | 1910±140 |
| DQN($n = 100$, $L = 10$)-EPSGREEDYEXPLORE | 2.203±0.012 | 1880±100 |

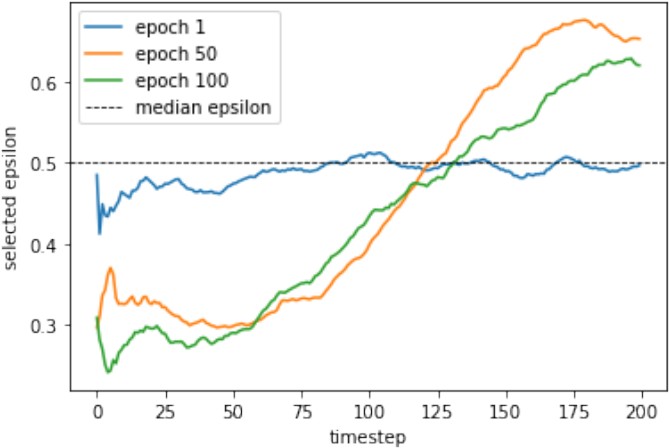

Figure 8: **Moving average evolution of the selected epsilon (from the best trace in Line 4 in Fig 5) across timesteps and epochs in the Acrobot environment (1 epoch = 200 timesteps).** In early epochs (epoch 0) when the guide is weak, selected epsilons are roughly uniform around the median, but as the guide is getting better (epoch 50), late timesteps require more exploration than first ones already learnt well by the guide. Eventually (epoch 100), the guide would further improve and less exploration is needed.

horizons a larger number of generated rollouts lead to a better performance. We also observed in our simulations that for the Acrobot to stay balanced, it was necessary (although not always sufficient) to have a reward larger than 2.6. We see from Fig 9 that this can be achieved with a simple agent such as RSACTOR but at the price of a very large number of generated rollouts. The goal is therefore to design a smarter agent that can come as close as possible to this performance with a limited budget.

## C.2  CARTPOLE

Fig 10 reports the performance obtained with RSACTOR and a CEM agent for different values of $n$ and $L$. For the CEM agent we run 5 iterations with the given $n$ and $L$ so the total budget is $5 \times n \times L$. A mean reward of 0.8 corresponds to a pole that succeeds at standing upright and being stable. We see that achieving such a performance requires a CEM agent and a planning horizon larger than $L = 100$.

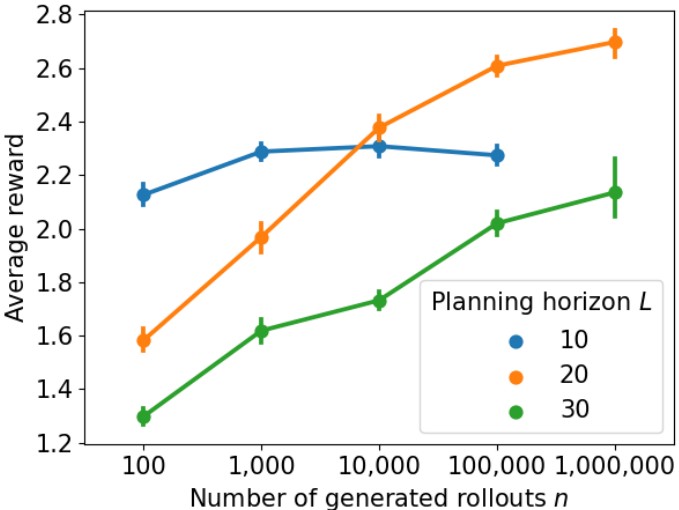

Figure 9: Performance obtained with RSAcTOR on the real Acrobot system for different planning horizons $L$ and number of generated rollouts $n$. The plot shows the mean rewards obtained for several randomly initialized episodes of 200 steps. The error bars give the associated 90% confidence intervals. Note that since Acrobot has a discrete action space with three actions, the total number of different rollouts for $h = 10$ is $n = 3^{10} = 59{,}049$. The performance shown for $h = 10$ and $n = 100{,}000$ thus only requires $n = 59{,}049$ rollouts.

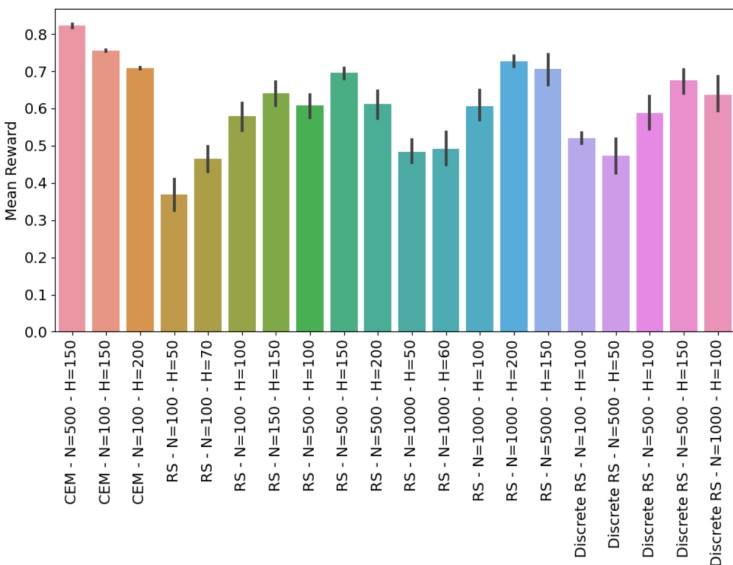

Figure 10: Performance obtained with RSAcTOR and CEM on the real Cartpole system for different planning horizons $L$ and number of generated rollouts $n$. The plot shows the mean rewards obtained for several randomly initialized episodes of 1000 steps. The error bars give the associated 90% confidence intervals.

