# OpenReview forum: "The guide and the explorer: smart agents for resource-limited iterated batch reinforcement learning"
_ICLR.cc/2023/Conference — Submitted to ICLR 2023_

### Official Review · Reviewer_F5c7 · 2022-10-24

**Confidence:** 2
**Correctness:** 3
**Technical Novelty And Significance:** 3
**Empirical Novelty And Significance:** 3
**Recommendation:** 6

**Clarity, Quality, Novelty And Reproducibility:**

Authors' motivation and the problem being addressed are clearly explained. However, the novelty of the proposed strategy is not clearly presented, nor its main components. I suggest to improved the text of sub section 3.4
Given that two standard problems are addressed (Acrobot & Cartpole Swing-up) the results are reproducible.

**Details Of Ethics Concerns:**

No concerns.

**Strength And Weaknesses:**

Strengths:
- Motivation
- Relevance of the problem being addressed

Weaknesses:
- The novelty of the proposed strategy needs to be better explained. I suggest to improve sub section 3.4
- The two problems used for the validation are rather simple. I recommend to use a much more complex problem to validate the proposes strategy.

**Summary Of The Paper:**

This paper addresses the  iterated/growing batch RL problem. Authors proposes a Dyna-Style Guide & Explore strategy, which most novel component seems to be an explorer (heating explorer) based on a temperature parameter. The proposed strategy is validated in two classical simple problems: Acrobot (discrete actions) and Cartpole Swing-Up (continuous actions).


**Summary Of The Review:**

In the context of iterated/growing batch RL, authors propose a Dyna-Style Guide & Explore strategy, which most novel components seems to be an explorer (heating explorer) based on a temperature parameter. The proposed strategy is validated in two simple, simulated problems: Acrobot (discrete actions) and Cartpole Swing-Up (continuous actions). The simplicity of these problems do not allow to assess properly the relevance of the proposed approach.

---

> ### Author Response · Authors · 2022-11-18
> **Response**
>
> Thank you for your review. All your comments are addressed in the general response.

---

### Official Review · Reviewer_PxDX · 2022-10-25

**Confidence:** 3
**Correctness:** 3
**Technical Novelty And Significance:** 3
**Empirical Novelty And Significance:** 2
**Recommendation:** 5

**Clarity, Quality, Novelty And Reproducibility:**

The paper is written well in general, but the presentation might need some improvement. The introduction is well motivated with practical examples. The technical details and connection between different components are clearly mentioned. The experimental results section is easy to follow, except some minor issues (e.g., bold numbers in Table 1 do not seem to the best ones).
The related work section looks exhaustive and solid to me (albeit some recent work on how exploration or evaluation is done in offline RL is missing). Some additional minor comments:
1.	Some guidance is needed on how an expert would pick the values of input parameters L, n.
2.	Details of hyperparameter tuning, e.g., temperature, T_i and probabilities, \epsilon_i would be useful.
3.	In experiments, how a model-free RL is trained to find the guide policy?


**Strength And Weaknesses:**

The paper extends the research on combining model-based and model-free RL methods for improving sample efficiency and resource utilization. Combining model-free RL with MPC is a well-studied problem but exploring this in an iterated batch RL setting is the key novelty of this paper. This is an extension of Kegl et. al. (2021), where rather than using simple random shooting, a Dyna-style approach is used to learn a dynamic model from gathered data that is used to collect future imaginary traces. Experimental results support the claim that such Dyna-style approach with efficient exploration improves the performance over random shooting.
While I appreciate the approach of using sophisticated Dyna-style approach for generating future traces, I have some reservations regarding the contributions of the paper. Firstly, both Dyna-style approach, and using the concept of model-based exploration are well-studied; this paper just used these ideas in the context of iterated batch RL setting. The paper lacks any theoretical results that can demonstrate the efficacy (in terms of improving two evaluation metrics) of the proposed models in a generic setting. The experiments are conducted on synthetic Acrobat and Cartpole problems, so without theoretical guarantees, it is not clear to me whether the claims are generalizable to other real-world offline RL problems.

**Summary Of The Paper:**

The paper proposed an efficient guide and exploration strategy to improve the performance of iterated batch RL. A Dyna-style explore, and guide method is used where an exploration model is learnt first using a model-based guiding approach that is then used to collect future trajectories. An end-to-end algorithm is proposed that learns a model free guide policy, learn exploration policy, and then collect optimized trajectories. Experimental results on Acrobat and Cartpole tasks demonstrate that the proposed method outperforms existing methods like DQN, SAC and RSActor.

**Summary Of The Review:**

Iterative batch RL is an interesting and growing research area that can make significant impact in adoption of RL. The ideas presented in the paper are interesting and novel in the sense that an efficient exploration with a learned guiding model can accelerate performance. Experiments demonstrate performance improvement in Acrobat and Cartpole environment. It would be really appreciated if some theoretical results were presented in terms of performance vs. organizational time constraints. Experiments on other real-world domains will further ground the claims. Adding ablation studies with different parameters (e.g., L, n) would be great as well, as my main concern is that choosing an optimal value for these parameters would be challenging.

---

> ### Author Response · Authors · 2022-11-18
> **Response**
>
> Thank you for your review.
>
> **Some guidance is needed on how an expert would pick the values of input parameters L, n. Choosing an optimal value for these parameters would be challenging**:
> The maximum budget $n \times L$ is imposed by the practical constraints related to the real application: the resources and time available for decision-time planning. Once this maximum budget has been defined, for the design of the algorithm one also needs to be aware that the choice of $L$ will depend on the maximum number of rollouts $n$ that can be associated to it as a larger value of $L$ increases the dimension of the search space. Finally model errors accumulate with the horizon $L$. While automatically selecting the best planning horizon $L$ is an important problem we consider that it is out of the scope for this paper.
>
> **Choice of the temperature set and details of the hyperparameter tuning, e.g., temperature, $T_i$, and probabilities, $\epsilon_i$, would be useful. How a model-free RL is trained to find the guide policy?**: Details are given in appendix A. While the user still has to define a set of temperatures and a set of probabilities, we believe that this is less error prone and more robust than considering a fixed exploration amount (such as adding a Gaussian noise with constant sigma).

---

> > ### Comment · Reviewer_PxDX · 2022-11-22
> > **Scores unchanged**
> >
> > Thank you for clarifying some of the questions. For me, the concerns regarding the lack of theoretical results and experiments on real-world examples still remain the same. So, I would keep my scores unchanged.

---

### Official Review · Reviewer_qJN2 · 2022-10-26

**Confidence:** 3
**Correctness:** 3
**Technical Novelty And Significance:** 2
**Empirical Novelty And Significance:** 2
**Recommendation:** 3

**Clarity, Quality, Novelty And Reproducibility:**

This paper is well written and has interesting ideas and insights but I do not believe ICLR is the right venue it in its current form. The contributions listed in subsection 1.1 are very vague and high-level compared to other works that appear in ICLR. Guide&Explore is the main novel algorithm presented, but the classical annealing algorithm it uses, augmented with a temperature set rather than a single temperature, does not seem significantly novel or well justified -- while there is discussion on how guide policies tend to underexplore, there is no theory based justification or discussion on why this type of heating method is the right change to make. In fact, it seems like learning the right set of temperatures to use for a given problem would require hyperparameter tuning and ultimately be very sample inefficient -- methods to identify the temperature set should be discussed in the paper. It is also unclear what specific kinds of problems this paper is specifically trying to address; the case studies on cartpole and acrobot seem vastly different than the types of use cases discussed initially. I think the missing piece is a rigorous link between the complexities discussed in section 1, the model formulation in 3.3, and the experimental setup in section 4.

I believe this paper requires significant revisions to rigorously define and characterize the specific classes of problems it is trying to address and to create a mathematical formulation that provides hard insights into the challenges of the problems considered. With this, the experiments could be better justified as being well suited for evaluating algorithms in such tasks. The paper also requires a more rigorous comparison to relevant state-of-the-art methods beyond DQN and SAC. For example, after reading the introduction I was expecting a comparison against imitation learning methods, since the premise seems to focus on problems wherein a human operator had spent significant time performing the task and thus there would be a huge amount of data to learn from.

As an aside, I think the telecommunications engineer example in the beginning is entertaining and relatable but could be greatly abbreviated for this short conference-style format.

The paper includes a reproducibility statement that suggests the results will be easily reproducible with released code, which is well appreciated.

**Strength And Weaknesses:**

# Strengths
## Novelty
- The heating explore based on using a set of different temperatures to create more exploratory policies seems novel and effective
## Significance
- The general classes of real-world problems considered by this paper are widespread, and this is an interesting investigation into using deep reinforcement learning to tackle them
## Clarity
- The algorithms and results figures are clear and informative
- The text is well edited and makes good use of references to prior works; both of these are well appreciated

# Weaknesses
## Novelty
- Novel algorithms and/or algorithmic changes are not clearly marked, making it harder to identify what is novel
## Significance
- It is hard to see how performance on cartpole and acrobot relate at all to the motivating example of a telecommunications engineer; cartpole and acrobot do not seem to reflect the considerations listed in section 1
- This approach seems to rely on determining the optimal temperature set for exploration -- it is not clear how one could feasibly identify a temperature set to use for novel problems
- There are no new theorems or proofs presented in this paper to rigorously explain why these problems are difficult or unsuited to other deep learning methods, or why the proposed methods are better
## Clarity
- The problems discussed in the first half of the paper at a high level and in vague terms, making it unclear what this paper is precisely trying to contribute or how its insights can/should be used by future works

**Summary Of The Paper:**

This paper explores approaches to iterative batch reinforcement learning, whereby relatively low-dimensional data is collected from physical systems and relatively simple actions/decisions must be made in response to these observations. This includes consideration of how the agent may only be given limited real-world decision making experience, but potentially more access to off-policy data. The paper shows that a novel "Guide&Explore" strategy achieves better performance than the baselines on two standard tasks: acrobot and cartpole.

**Summary Of The Review:**

While the paper presents an interesting look at using and modifying RL methods for iterated batch RL problems that have some similarities to real world intelligent control problems, it does not yet have the precise and rigorous problem setup and mathematical theory-backed analysis expected for ICLR papers in this area. It could also use more experimental results ideally exploring more diverse test problems and more state-of-the-art baselines (e.g. in imitation learning and other fields).

---

> ### Author Response · Authors · 2022-11-18
> **Response**
>
> Thank you for your review.
>
> **It is hard to see how performance on cartpole and acrobot relate at all to the motivating example of a telecommunications engineer; cartpole and acrobot do not seem to reflect the considerations listed in section 1**:
> We use Acrobot and Cartpole to show the performance of our method on well-known open sourced benchmarks on which we can compare to previous state-of-the-art results. The way we evaluate our method on these two systems is made to mimick the constraints and specificities listed in Section 1.
>
> **This approach seems to rely on determining the optimal temperature set for exploration -- it is not clear how one could feasibly identify a temperature set to use for novel problems**:
> The set of temperatures has indeed to be determined by the user. We believe that it is however less error prone and more robust to select a set of temperatures than to rely on one fixed exploration amount.
>
> **Comparison against imitation learning methods**:
> It is often possible to convince the system engineers to deploy a (safe) random exploration policy for a few trials and then improve on this policy. We therefore considered an iterated batch setting in our experiments where we start from a random policy. Studying the case where we would start from an expert dataset (collected by a policy designed by engineers) is of great interest and we defer it to future work.

---

> > ### Comment · Reviewer_qJN2 · 2022-11-22
> > **Thank you for the response**
> >
> > I appreciate the clarifications, although I have decided to keep my original score.
> >
> > Regarding the first point, in a future version of this paper I'd like to see a little bit more discussion on that mapping from acrobot and cartpole to real-world problems. One thing might be to formalize the variables of interest in the intro, work them into the discussion on related work, and then revisit them in the experimental setup. At the moment it feels like the thread linking the paper sections together is a little too thin.
> >
> > Regarding the second point, I'm not convinced the belief that it is "less error prone and more robust to select a set of temperatures than to rely on one fixed exploration amount" is correct, and would need to see an efficient strategy to pick these temperatures and evidence that it is advantageous relative to the best strategy for choosing a single temperature.
> >
> > Regarding the third point, I'd first note that my main concern was the low number of baselines. I do not think that comparing against vanilla DQN/SAC or a random actor is sufficient -- there are other algorithms that can be used to guide exploration and improve training efficiency and should be compared against. Second, regarding the suggestion of imitation learning, I'm still unsure why it's not feasible/informative to compare against these approaches as a baseline. It seems like they could be easily trained using the data collected between "system access" steps. It also seems well aligned with the original motivating case of an engineer who has been performing some task for a long time and wishes to automate it; it seems it would be trivial to have an expert dataset by that point. In any case, my main point is that the paper needs at least 1 or 2 more sophisticated baselines (specifically, I'd like to see alternative guides).

---

> > > ### Author Response · Authors · 2022-11-28
> > > **Thank you very much for the suggestions and feedbacks**
> > >
> > > Thank you very much for your feedback and suggestions. We will definitely consider them to improve our paper.
> > >
> > > For the selection of the temperatures, there is, to the best of our knowledge, no strategy to pick a single temperature besides running an hyperparameter optimization with evaluations on the real system, which would also require a set or interval of temperatures. Our strategy performs this optimization while we are planning with the model. Furthermore, Figure 8 in the appendix shows that the optimal temperature can also depend on how the guide performs at a given state: if the guide is good then we can follow its decision and a low temperature is good whereas when the guide is weak it is better to explore with a high temperature. Using a single temperature would not be able to adapt to the guide's performance.

---

### Official Review · Reviewer_fuzN · 2022-10-31

**Confidence:** 3
**Correctness:** 3
**Technical Novelty And Significance:** 2
**Empirical Novelty And Significance:** 2
**Recommendation:** 3

**Clarity, Quality, Novelty And Reproducibility:**

The paper is overall clearly written and easy to follow. The experiments are conducted using simple agents in simple environments, so I think it is likely that the experiments can be reproduced.

**Strength And Weaknesses:**

Strength:

The paper is overall well-written and studies an important problem.

Weakness:

The paper is very similar to the AlphaZero algorithm by Silver et al. (2017), therefore, I don't think this paper has enough novelty.
The guide policy is similar to the prior policy in AlphaZero as the authors mentioned in Section 3.2. Instead of learning the guide during MCTS training, this paper learns the guide using simpler model-free RL algorithms.
The decision-time planning is also similar to AlphaZero. In fact, AlphaZero uses a more advanced MCTS technique whereas this paper uses a simple rollout procedure and chooses the action that leads to the best return. In other words, this paper only uses one step of MCTS.
Therefore, I think this paper is a simple modification of the AlphaZero algorithm and thus it is not novel enough as an ICLR paper.

**Summary Of The Paper:**

This algorithm proposes an algorithm for iterated batch reinforcement learning. The algorithm uses model-free RL to learn a guide policy, and then uses decision-time planning to improve the policy. The decision-time planning uses some exploration method and a rollout procedure to get a good action.

**Summary Of The Review:**

The paper lacks novelty since it is very similar to AlphaZero, and in fact, it is almost a simplified version of AlphaZero. The experiments are performed only in simple environments. Therefore, I don't think this paper makes a significant contribution to the community.

---

> ### Author Response · Authors · 2022-11-18
> **Response**
>
> Thank you for your review.
>
> **Comparison with AlphaZero**:
> We are well aware of the AlphaZero paper (Silver et al. 2017), and although we share some common principles with this algorithm, our approach is still based on significant differences that we tried to explain in Section 2:
> - First, AlphaZero is a model-free approach and relies on many system access steps (typically millions vs thousands in MBRL). It is clearly not a feasible direction for the type of real-life systems we aim to solve. Perhaps the reviewer was referring to Muzero (Schrittwieser et al. 2020), a model-based adaptation of AlphaZero. Yet, it learns a latent state-space model which is trained jointly with the representation and actor/critic functions. We wanted to uncouple these components by proposing a simplified approach that wisely leverages different (already existing) components, yet achieves state-of-the-art performance. We believe that such decoupling is a must if the goal is not purely engineering (performance) but also scientific (understanding what matters and why). Reasoning in the natural state-space is also of interest for real-life applications where a practitioner would want to separately train and evaluate the model or the agent, use it for other purposes, or possibly correct it with expert knowledge.
> - Muzero was initially designed for environments with discrete actions. Although an interesting adaptation was proposed (Sampled Muzero, Hubert et al. 2021) to deal with continuous actions, our approach works effortlessly in both settings.
> - Our decision-time planning exhibits important differences: Muzero relies on a Monte-Carlo tree search which is in essence built by sequentially exploring the tree (as each tree search updates rules for the next one) with a priori unknown action sequence horizon. This results in an irreducibly slow planning at test-time, which is again not desired for real-life applications. Thus, our approach is not equivalent to "one step of MCTS", but more correctly to a fixed horizon parallel evaluation of multiple action sequences (multiple parallel tree searches with 'frozen' rules), considerably accelerating the planning.
> - In spirit, the UCB constants and the count-based measure play the same role as our prior temperature set as they govern how far to diverge from the prior policy in order to explore. Yet, to our knowledge, these constants need very fine adjustment and are crucial for the algorithm performance, while our simple temperatures set prior presents more robustness to its tuning as we observed in our experiments: minor alterations to its range or growth scheme do not degrade performance.
> - The method we suggest is easy to implement and debug. It relies on model-free algorithms that are available in popular and well-maintained libraries such as stable-baselines3. The other components (decision-time planning, model learning) are also easy to implement.

---

> > ### Comment · Reviewer_fuzN · 2022-11-20
> > **Thanks for your response**
> >
> > I still think this paper is quite similar to AlphaZero and thus the contribution is incremental. I decided to keep my score.

---

### Author Response · Authors · 2022-11-18
**To all the reviewers**

We thank the reviewers for their time spent reviewing our paper and their constructive feedbacks. We first address general comments and then reply individually to the comments that are specific to each reviewer.

**Novelty**:
We believe that even though most of the components used in our Guide\&Explore strategy are known, the study of their combination and the importance of each of the components to reach state-of-the-art performance is novel and highly relevant for the community. Furthermore integrating an adaptive exploration scheme while planning is a novel idea that avoids setting a fixed noise quantity (as done for instance in MBPO (Argenson and Dulac-Arnold, 2021)). This component is necessary to achieve state-of-the-art performance and can adapt to the performance of the guide at a given state (see Figure 8 in the paper).

**Lack of theory**:
Although we did not integrate a theoretical study of our algorithm, our study shows the impact of the different components of our algorithm (guide, decision-time planning, exploration) and we believe that our results will be useful for the community and the practitioners.

**Only Cartpole and Acrobot**:
Considering two environments allows to run an extensive ablation study. We also remind, as argumented in the section 4.1, that Acrobot is a not so simple environment which requires expensive decision-time planning.

---

### Decision · Program_Chairs · 2023-01-20

**Decision:**

Reject

**Justification For Why Not Higher Score:**

While all reviewers felt the paper tackled an important problem and that the approach was well-motivated, they had concerns about the novelty of the approach and the sufficiency of the evaluation. On the novelty side, the method is a new combination of existing parts (Dyna, rollouts, expert iteration) and is similar to a number of existing algorithms in the literature. This is not in principle a problem, if the paper can demonstrate that this particular combination works better or is more well-motivated than existing approaches. However, the paper does not compare to any other related approaches---the comparisons are mostly to ablations rather than true baselines. I appreciate the authors' attempt to clarify some of the differences between their approach and others in their rebuttal, but believe this needs to be done empirically, not just via argument. On the evaluation side, in addition to the lack of appropriate baselines, the reviewers felt that the domains were too simple and that they were not representative of the real-world problems presented in the introduction of the paper, making it difficult to evaluate how well the proposed method would actually fare in difficult real-world settings.

**Justification For Why Not Lower Score:**

N/A

**Metareview: Summary, Strengths And Weaknesses:**

This paper proposes a model-based method for iterated batch RL in which an agent learns a model of the transition function which it uses within search to improve a base policy. In comparison to existing Dyna-style approaches, the proposed method uses decision-time planning when selecting actions; in comparison to pure MPC, it learns a guide policy using model-free techniques. The method is evaluated in two domains, Acrobot and Cartpole Swing-Up, showing that it outperforms pure Dyna, MPC, and model-free baselines.

While all reviewers felt the paper tackled an important problem and that the approach was well-motivated, they had concerns about the novelty of the approach and the sufficiency of the evaluation. On the novelty side, the method is a new combination of existing parts (Dyna, rollouts, expert iteration) and is similar to a number of existing algorithms in the literature. This is not in principle a problem, if the paper can demonstrate that this particular combination works better or is more well-motivated than existing approaches. However, the paper does not compare to any other related approaches (e.g. MuZero [1], EfficientZero [2], Policy Gradient Search [3], SAVE [4], etc.)---the comparisons are mostly to ablations rather than true baselines. I appreciate the authors' attempt to clarify some of the differences between their approach and others in their rebuttal, but believe this needs to be done empirically, not just via argument. On the evaluation side, in addition to the lack of appropriate baselines, the reviewers felt that the domains were too simple and that they were not representative of the real-world problems presented in the introduction of the paper, making it difficult to evaluate how well the proposed method would actually fare in difficult real-world settings.

In light of the concerns raised by the reviewers, I do not feel this paper is ready for publication at ICLR and recommend rejection. I encourage the authors to revise the paper to clarify the novelty of their contributions with respect to related approaches (by including them as baselines) and to perform experiments on domains that are more representative for the real-world iterated batch setting.

1. Schrittwieser, J., Antonoglou, I., Hubert, T., Simonyan, K., Sifre, L., Schmitt, S., ... & Silver, D. (2020). Mastering atari, go, chess and shogi by planning with a learned model. Nature, 588(7839), 604-609.
2. Ye, W., Liu, S., Kurutach, T., Abbeel, P., & Gao, Y. (2021). Mastering atari games with limited data. Advances in Neural Information Processing Systems, 34, 25476-25488.
3. Anthony, T., Nishihara, R., Moritz, P., Salimans, T., & Schulman, J. (2019). Policy gradient search: Online planning and expert iteration without search trees
4. Hamrick, J. B., Bapst, V., Sanchez-Gonzalez, A., Pfaff, T., Weber, T., Buesing, L., & Battaglia, P. W. (2019). Combining q-learning and search with amortized value estimates.

**Summary Of Ac-Reviewer Meeting:**

N/A